# Antiretroviral therapy initiation and outcomes of hospitalized HIV-infected patients in Uganda—An evaluation of the HIV test and treat strategy

**Andrew Katende**[1,2]*, **Lydia Nakiyingi**[1,3], **Irene Andia-Biraro**[1,4,5], **Thomas Katairo**[6], **Richard Muhumuza**[4], **Andrew S. Ssemata**[4], **Christopher Nsereko**[7], **Fred C. Semitala**[1,6,8], **David B. Meya**[1,3]

1 Department of Medicine, School of Medicine, College of Health Sciences, Makerere University, Kampala, Uganda, 2 Ifakara Health Institute, Ifakara, Tanzania, 3 Infectious Diseases Institute, College of Health Sciences, Makerere University, Kampala, Uganda, 4 Medical Research Council/Uganda Virus Research Institute and London School of Hygiene and Tropical Medicine Uganda Research Unit, Entebbe, Uganda, 5 Department of Clinical Research, Faculty of Infectious and Tropical Disease, London School of Hygiene and Tropical Medicine, London, United Kingdom, 6 Infectious Disease Research Collaboration, Kampala, Uganda, 7 Entebbe Regional Referral Hospital, Entebbe, Uganda, 8 Makerere University Joint AIDS Program (MJAP), Kampala, Uganda

* katendeandrew13@gmail.com

**Data Availability Statement:** Due to the small sample size and the detailed content of the interview transcripts, including sensitive and

## Abstract

### Background

Uganda adopted the HIV Test and Treat in 2016. There is paucity of data about its implementation among hospitalized patients. We aimed to determine the proportion of patients initiating anti-retroviral therapy (ART) during hospitalization, barriers and mortality outcome.

### Methods

In this mixed methods cohort study, we enrolled hospitalized patients with a recent HIV diagnosis from three public hospitals in Uganda. We collected data on clinical characteristics, ART initiation and reasons for failure to initiate ART, as well as 30 day outcomes. Healthcare workers in-depth interviews were also conducted and data analyzed by sub-themes.

### Results

We enrolled 234 patients; females 140/234 (59.8%), median age 34.5 years (IQR 29–42), 195/234 (83.7%) had WHO HIV stage 3 or 4, and 74/116 (63.8%) had CD4 $\leq$ 200 cell/μL. The proportion who initiated ART during hospitalization was 123/234 (52.6%) (95% CI 46.0–59.1), of these 35/123 (28.5%) initiated ART on the same day of hospitalization, while 99/123 (80.5%) within a week of hospitalization. By 30 days 34/234 (14.5%) (95% CI 10.3–19.7) died. Patients residing $\geq$ 35 kilometers from the hospital were more likely not to initiate ART during hospitalization, [aRR = 1.39, (95% CI 1.22–1.59). Inadequate patient preparation for ART initiation and advanced HIV disease were highlighted as barriers of ART initiation during hospitalization.

potentially identifying information. The Makerere University, school of Medicne research ethics committee does not approve public release of this type of data. However data supporting the findings of this study will be available upon reasonable request to the the corresponding author. All relevant quantitative data are within the manuscript and its Supporting information files. A quantitative data stata data set has been uploaded as Supporting information. Requests for Qualitative data can be sent to the Makerere School of Medicine Research Ethical Committee, rresearch9@gmail.com or research@chs.mak.ac.ug.

**Funding:** This work is supported by the Fogarty International Center of the National Institutes of Health under Award Number D43 TW010037. The content is solely the responsibility of the authors and does not necessarily represent the official views of the National Institutes of Health.

**Competing interests:** The authors have declared that no competing interests exist.

## Conclusion

In this high HIV prevalence setting, only half of newly diagnosed HIV patients are initiated on ART during hospitalization. Inadequate pre-ART patient preparation and advanced HIV are barriers to rapid ART initiation among hospitalized patients in public hospitals.

## Introduction

Early ART initiation among people living with HIV (PLHIV) reduces rates of HIV transmission, morbidity and mortality [1–3]. In 2015, the World Health Organization (WHO) recommended the HIV test-and-treat strategy where immediate ART should be offered to all newly diagnosed PLHIV irrespective of CD4 count [4], similary in 2016 Uganda adopted this strategy [5,6].

The clinical decision to initiate ART among hospitalized patients can be challenging, especially in settings with limited diagnostic capacity to rule out opportunistic infections. Hospitalized patients are commonly symptomatic, more likely to have co-morbidities and organ dysfunction which may complicate immediate ART initiation [7,8]. The unexpected HIV diagnosis during hospitalization causes psychological stress and denial, which might affect the patients' readiness to accept ART during hospitalization [9]. A cross-sectional study conducted at the National Referral Hospital in Kampala when Ugandan HIV guidelines recommended ART initiation below a CD4 threshold of 200 cells/μL demonstrated that there was a delay to initiate ART among 75% of hospitalized ART-eligible patients [10]. This study further showed that one third of PLHIV died within two weeks. Previous studies have mainly focused on ambulatory patients in outpatient clinics, community-based ART initiation, and key populations to demonstrate the feasibility and benefits of HIV test-and-treat [11,12]. Four years since Uganda adopted the HIV test-and-treat strategy, there remains a paucity of data on ART initiation among hospitalized patients. We therefore, aimed to determine the proportion of public hospital patients initiating ART during hospitalization, their outcomes, and barriers to in-patient ART initiation.

## Methods

### Study design and setting

Between December 2019 and March 2020, we conducted a mixed methods prospective cohort study in which we enrolled HIV infected patients and healthcare workers (HCW) from three public hospitals in Uganda, namely; Kiruddu Referral hospital (Kiruddu hospital), Entebbe Regional Referral Hospital (Entebbe Hospital) and China-Uganda Friendship Hospital Naguru (Naguru hospital). Kiruddu hospital is a 200-bed capacity hospital located ten kilometers from the city center in Kampala. The hospital has an HIV clinic that manages 300 patients weekly and admits approximately 10 patients with recently diagnosed HIV infection weekly. Entebbe Hospital is a 200-bed capacity hospital located approximately 35 kilometers from Kampala city and has an HIV clinic that cares for almost 200 patients daily. Approximately 6 patients with recently diagnosed HIV infection are hospitalized weekly. Naguru hospital is a 100-bed capacity hospital, located in Kampala and has an outpatient HIV clinic with 300 patients managed daily and about 6 patients hospitalized with recently diagnosed HIV infection.

## Study population and participant recruitment

We screened hospitalized patients with a recent HIV diagnosis based on routine hospital HIV screening per national guidelines. Patients were eligible if they were; 12 years or older, ART naïve, hospitalized for ≥24 hours and gave informed consent/ assent to participate in the study. Patients were excluded if they did not have a personal phone or contacts of a next-of-kin to enable 30 day- follow up.

At enrolment, participants were interviewed to obtain information on socio-demographic and clinical data. A follow up interview on the day of discharge or death was used to collect data on ART status and dates of ART initiation. This information was obtained from the patients' hospital chart, ART card documents and ART pill counts. The 30-day follow up visit was performed via phone call, 30 days from the day of admission. A participant was considered lost to follow-up if they were un-contactable both physically and through daily phone calls over a period of one week. We also conducted in-depth interviews among twelve purposively selected HCW from the three hospitals these included Internal Medicine specialist trainees (n = 4), intern doctors (n = 4), ART nursing officers (n = 2) and HIV counselors (n = 2).

## Data management and analysis

**Quantitative analysis.** A pre-tested questionnaire was used to collect socio-demographic (sex, age, marital status, occupation, education status, residence and monthly income) and clinical data (WHO HIV stage, CD4, ALT, AST, creatinine, hemoglobin) through patient interviews and hospital chart review. Data was entered in Epi-Data Version 4.6 (Denmark) and then exported to STATA software V15.0 (Stata Corp, College Station, TX) for analysis S1 File. Continuous variables were summarized into means and medians. Categorical variables were summarized and tabulated as frequencies and percentages.

The proportion (95% confidence interval) of hospitalized patients with a new HIV diagnosis who initiated ART during hospitalization was calculated as the number of patients initiated on ART out of the total number of patients recruited in the study.

We adopted the modified Poisson regression model with clustered standard errors (clustered at hospital levels) to predict risk ratios (RR). All variables that had $p$-values $< 0.2$ from bivariate analysis were considered for multivariate analysis. In the multivariate analysis, variables were considered using backward stepwise method. Variables with a $>10\%$ difference between the crude and adjusted prevalence ratios were considered confounders. All variables were considered to be significant when the $p$-value $<0.05$.

**Qualitative analysis.** In-depth interviews using an interview guide lasting between 40–60 minutes were conducted before the quantitative data collection process, all interviews were audio recorded and transcribed verbatim to generate transcripts used in the analysis S2 File. Data analysis started by developing a codebook and scanning through the transcripts to generate codes, sub themes and broad themes. Transcripts review and codebook development was performed in parallel with the data collection. The analysis was accomplished by the first author (AK) together with two social scientists (RM and ASS) by coding the data manually to the point of saturation, where no novel additional information was being collected. Data was analyzed following the thematic analysis approach that allows analytical themes to emerge from the data and to define codes during the process of reading, exploration and coding responses [13,14]. The broad themes represented the main thematic areas that included: (a) understanding of HIV test-and-treat policy, (b) conditions to start ART as per HIV test-and-treat policy, (c) when a newly diagnosed individual should start ART, (d) reasons for not starting ART immediately, (e) factors associated with failure to start ART, (f) facilitators of early

ART initiation, (g) challenges in delivering HIV test-and-treat in hospitals and, (h) solutions to the listed problems.

**Ethical considerations.** This study was approved by the Scientific Review Committee, Department of internal medicine, Makerere University College of Health Sciences. The study was also approved by the School of Medicine Research and Ethics Committee (REC REF 2019–165). It also received clearance from all hospital sites. Written Informed consent was sought and obtained from all participants.

## Results

We screened 250 hospitalized patients with a recent HIV diagnosis at three public hospitals of which 234 were found eligible and enrolled in the study, (Fig 1). Of the 234 participants, 120/234 (51.3%) were recruited from Kiruddu hospital, 59/234 (25.2%) from Naguru hospital and 55/234 (23.5%) from Entebbe hospital.

### Socio-demographic and clinical characteristics of hospitalized patients with a recent HIV diagnosis

Among the 234 participants, 59.8% were female and the median age was 34.5 years, (IQR 29–42). The majority (93.2%) had at least primary education and were involved in an income-generating activity. More than a third (32%) had a monthly income of less than 26 USD. The majority of participants from Kiruddu Hospital resided more than 35 kilometers from the hospital Table 1.

The majority of participants had WHO stage 3 or 4 HIV disease. Half of the participants had a documented CD4 cell count, 116/234 (49.6%) of which 48/116 (41.4%) had CD4 <100 cells/μL. Attending clinicians made a diagnosis of either Tuberculosis in 36/234 (15.4%) or Cryptococcal meningitis in 14/234 (5.9%) of the participants. Almost one-third of the participants had severe anemia (hemoglobin < 8g/dl) Table 2.

### ART initiation during hospitalization and 30-day outcomes

The median time from admission to HIV testing was 1 day (IQR 1–2) and the median time of hospitalization was 5 days (IQR 3–7). The proportion of participants who initiated ART during hospitalization was 123/234, 52.6%, (95% CI 46.0–59.1) and the median time from admission to ART initiation was 4 days (IQR 1–11). Cumulatively 35/123, 28.5%, (95% CI 20.7–37.3) had initiated ART by day one of hospitalization; 99/123, 80.5%, (95% CI 72.4–87.1) had initiated by day 7 and 113/123, 91.9%, (95% CI 85.6–96) by day 14.

After 30 days of follow up, 19/234 (8.1%) were lost to follow up, 34/234 (14.5%) (95% CI 10.3–19.7) died, 30/234 (12.8%) more patients had initiated ART since discharge, making the overall proportion of participants on ART after 30 days 146/215, (67.9%). Over two-thirds of deaths 22/34 (64.7%) occurred during hospitalization. Mortality among patients who did not initiate ART during hospitalization was significantly higher compared to those who started ART during hospitalization [32/110 29.1% (95% CI 20.8–38.5) versus 2/123, 1.6% (95% CI 20.6–38.2), $p$-value < 0.001)].

### Factors associated with failure to initiate ART during hospitalization

Among the 234 participants, 110 did not start ART during their stay in the hospital, 47%, (95% CI 40.5–53.6). Nearly a third were either too sick or died before starting ART, the main reason for not starting ART was being diagnosed with either Tuberculosis or cryptococcal meningitis. Approximately one in every five of the participants 25/110 (22.7%) opted not to start ART

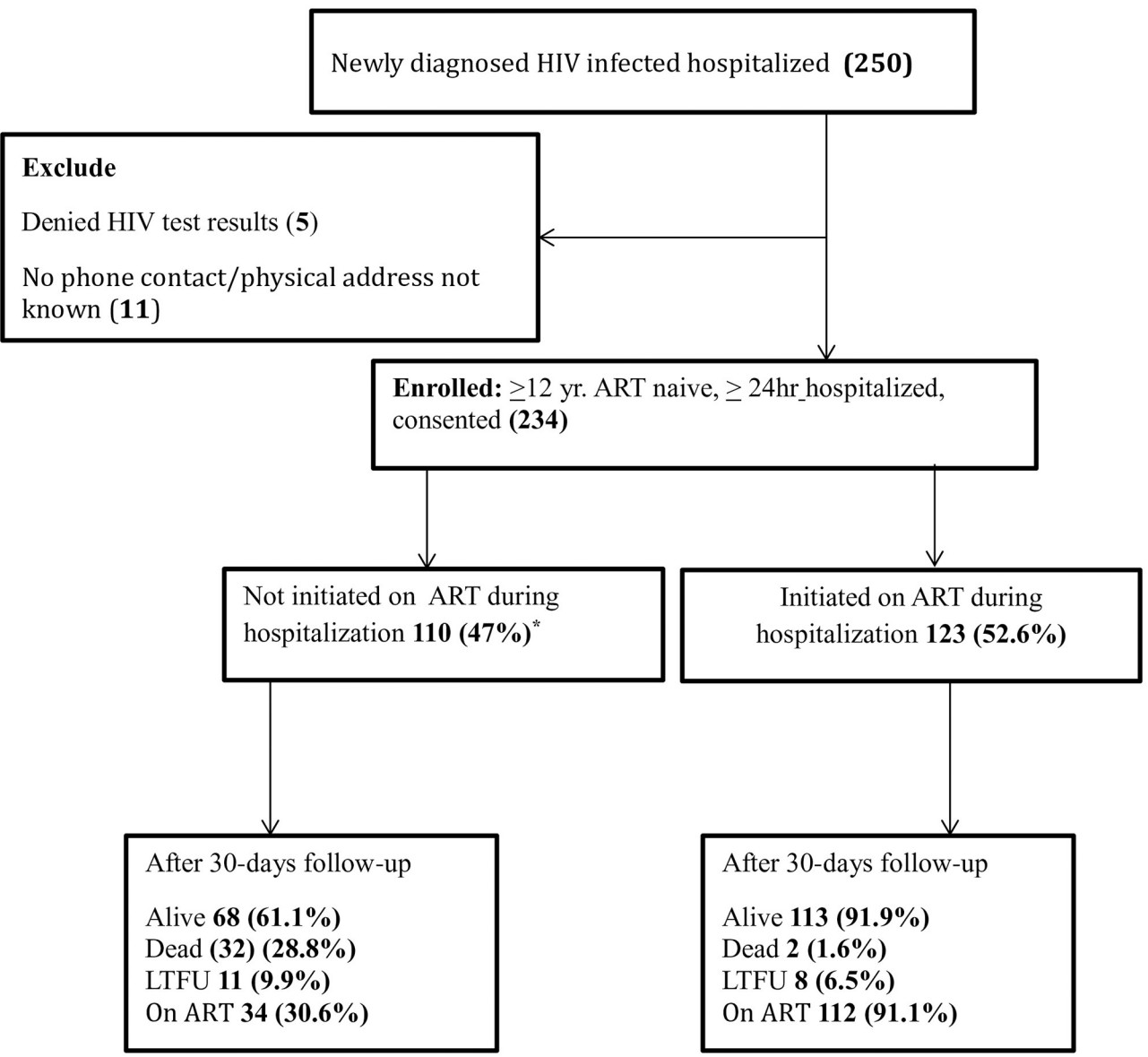

**Fig 1. Study profile of hospitalized patients with a recent HIV diagnosis.** * One patient was not assessed for ART status at discharge.

during hospitalization. Overall ART unavailability was the least common reason for not starting ART during hospitalization Table 3.

In the bivariate analysis, participants' distance from the hospital, education status, being on oxygen therapy, occupation status, and WHO HIV staging were were significantly associated with not initiating ART during hospitalization Table 4.

However, when included in the multivariate model, residing more than 35 km away from the hospital was a predictor for not starting ART during hospitalization compared to those residing within the 35 km, [aRR = 1.39, (95% CI 1.22–1.59); $p$-value <0.001]. Having atleast primary education, [aRR = 0.68, (95% CI 0.59, 0.78); $p$-value 0.001] and being on Oxgyen therapy, [aRR = 0.49, (95% CI 0.37, 0.65); $p$-value <0.001] were also predictors for not initiating ART during hospitalization.

**Table 1. Baseline demographic characteristics of hospitalized patients with a recent HIV diagnosis stratified by study site.**

| Variable | Overall (n = 234) | Kiruddu (N = 120) | Naguru (n = 59) | Entebbe (n = 55) |
|---|---|---|---|---|
| **Age** | | | | |
| <25 years | 26 (11.1) | 13 (10.8) | 3 (5.1) | 10 (18.2) |
| 25–49 | 181 (77.4) | 88 (73.3) | 48 (81.4) | 45 (81.8) |
| >50 years | 27 (11.5) | 19 (15.8) | 8 (13.6) | 0 (0.0) |
| **Sex** | | | | |
| Male | 94 (40.2) | 47 (39.2) | 30 (50.9) | 17 (30.9) |
| Female | 140 (59.8) | 73 (60.8) | 29 (49.2) | 38 (69.1) |
| Distance * | | | | |
| <35Km | 124 (53.0) | 33 (27.5) | 45 (76.3) | 46 (83.6) |
| >35Km | 110 (47.0) | 87 (72.5) | 14 (23.7) | 9 (16.4) |
| **Marital status** | | | | |
| Not married | 125 (53.4) | 78 (65.0) | 35 (59.3) | 12 (21.8) |
| Married | 109 (46.6) | 42 (35.0) | 24 (40.7) | 43 (73.2) |
| **Education status** | | | | |
| None | 16 (6.8) | 8 (6.7) | 0 (0.0) | 8 (14.6) |
| Primary | 95 (40.6) | 62 (51.7) | 10 (17.0) | 23 (41.8) |
| Secondary | 101 (43.2) | 43 (35.8) | 37 (62.7) | 21 (38.2) |
| Tertiary | 22 (9.4) | 7 (5.8) | 12 (20.3) | 3 (5.5) |
| **Occupation** | | | | |
| Employed | 182 (77.8) | 83 (69.2) | 50 (84.8) | 49 (89.1) |
| Not employed | 52 (22.2) | 37 (30.8) | 9 (15.3) | 6 (10.9) |

• Distance in Kilometers from the hospital.

**Table 2. Baseline clinical characteristics of hospitalized patients with a recent HIV diagnosis stratified by study site.**

| Variable | Overall (n = 234) | Kiruddu (N = 120) | Naguru (n = 59) | Entebbe (n = 55) |
|---|---|---|---|---|
| **Suspected TB or CM** | | | | |
| Yes | 119 (50.8) | 71 (59.2) | 25 (42.4) | 23 (41.8) |
| No | 115 (49.2) | 49 (40.8) | 34 (57.6) | 32 (58.2) |
| **Needed oxygen therapy** | | | | |
| Yes | 21 (9.0) | 6 (5.0) | 12 (20.3) | 3 (5.5) |
| No | 213 (91.0) | 114 (95.0) | 47 (79.7) | 52 (94.6) |
| **Able to self-feed** | | | | |
| Yes | 204 (87.2) | 99 (82.5) | 52 (88.1) | 53 (96.4) |
| No | 30 (12.8) | 21 (17.5) | 7 (11.9) | 2 (3.6) |
| **HIV WHO stage** | | | | |
| 1 to 2 | 38 (16.3) | 11 (9.2) | 11 (18.6) | 16 (29.1) |
| 3 to 4 | 195 (83.7) | 108 (90.8) | 48 (81.4) | 39 (70.9) |
| **CD4 (cell/mm$^3$)** | | | | |
| <200 | 74 (63.8) | 63 (76.8) | 0 (0.0) | 11 (32.4) |
| 200–499 | 36 (31.0) | 17 (20.7) | 0 (0.0) | 19 (55.9) |
| >500 | 6 (5.2) | 2 (2.4) | 0 (0.0) | 4 (11.8) |
| **Hb (g/dl)** | 10.0 (7.7, 11.0) | 9.9 (7.4, 11.0) | 10 (7.0, 12.0) | 10.0 (9.0, 11.0) |
| **Creatinine (µmol/L)** | 84 (61, 104) | 75 (52, 98.9) | 84 (69, 105) | 98 (87, 114) |

**Table 3. Reasons for not initiating ART during hospitalization stratified by study site.**

| Reason for not initiating ART | Overall (n = 110) | Kiruddu (n = 66) | Naguru (n = 28) | Entebbe (n = 16) |
|---|---|---|---|---|
| Confirmed TB/CM | 50 (45.5) | 38 (57.6) | 4 (14.2) | 8 (50.0) |
| ART unavailability | 1 (0.91) | 0 (0.0) | 1 (3.6) | 0 (0.0) |
| Opted to Start from a different ART clinic | 10 (9.1) | 3 (4.6) | 6 (21.4) | 1 (6.3) |
| Declined ART when offered | 15 (13.6) | 7 (10.6) | 6 (21.4) | 2 (12.5) |
| Died before starting ART | 22 (20.0) | 14 (21.2) | 4 (14.3) | 4 (25.0) |
| Too sick to start ART | 8 (7.3) | 2 (3.0) | 5 (17.9) | 1 (6.3) |
| No necessary lab workup | 4 (3.6) | 2 (3.0) | 2 (7.1) | 0 (0.0) |

**Table 4. Multivariate analysis for predictors of failure to initiate ART during hospitalization.**

| Variable | Started (n = 123) | Not started (n = 110)* | RR | P-value | aRR | 95% CI | P-value |
|---|---|---|---|---|---|---|---|
| Age | | | | | | | |
| <25 years | 11 (8.9) | 15 (13.6) | 1 | | | | |
| 25–49 | 100 (81.3) | 80 (72.7) | 0.77 | 0.083 | | | |
| >50 years | 12 (9.8) | 15 (13.6) | 0.96 | 0.905 | | | |
| Sex | | | | | | | |
| Male | 45 (36.6) | 48 (43.6) | 1 | | | | |
| Female | 78 (63.4) | 62 (56.4) | 0.86 | 0.095 | | | |
| Distance from hospital | | | | | | | |
| <35KM | 74 (60.2) | 50 (45.5) | 1 | | 1 | | |
| >35KM | 49 (39.8) | 60 (54.6) | 1.37 | <0.001 | 1.39 | 1.22, 1.59 | <0.001 |
| Marital status | | | | | | | |
| No partner | 64 (52.0) | 60 (54.6) | 1 | | | | |
| Partner | 59 (48.0) | 50 (45.5) | 0.95 | 0.599 | | | |
| Education status | | | | | | | |
| None | 5 (4.1) | 11 (10.0) | 1 | | 1 | | |
| Primary | 50 (40.7) | 45 (40.9) | 0.69 | <0.001 | 0.68 | 0.59, 0.78 | <0.001 |
| Secondary | 55 (44.7) | 46 (41.8) | 0.66 | <0.001 | 0.66 | 0.60, 0.73 | <0.001 |
| Tertiary | 13 (10.6) | 8 (7.3) | 0.55 | 0.074 | 0.57 | 0.31, 1.04 | 0.065 |
| Occupation | | | | | | | |
| Not employed | 25 (20.3) | 27 (24.6) | 1 | | | | |
| Employed | 98 (79.7) | 83 (75.5) | 0.88 | <0.001 | | | |
| Income | | | | | | | |
| <100000 | 35 (28.5) | 40 (36.4) | 1 | | | | |
| 100000–500000 | 83 (67.5) | 60 (54.6) | 0.79 | 0.12 | | | |
| >500000 | 5 (4.1) | 10 (9.1) | 1.25 | 0.279 | | | |
| WHO staging | | | | | | | |
| 1 to 2 | 33 (26,8) | 5 (4.6) | 1 | | | | |
| 3 to 4 | 90 (73.2) | 104 (95.4) | 4.07 | 0.001 | | | |
| Oxygen Therapy | | | | | | | |
| Yes | 3 (2.4) | 18 (16.4) | 1 | | 1 | | |
| No | 120 (97.6) | 92 (83.6) | 0.51 | <0.001 | 0.49 | 0.37, 0.65 | <0.001 |
| Baseline Hb (g/dl) | 10.0 (9.0, 11.4) | 8.9 (6.6, 11.0) | 0.9 | 0.149 | | | |

* one participant was not assessed for ART status hence N = 233.

## Qualitative findings: Barriers to initiating ART during hospitalization

The HCWs factors hindering ART initiation during hospitalization were categorized into; i) patient related, ii) healthcare worker related, and iii) system related factors.

## Patient related factors: Stigma, knowledge, and family support

All HCWs were certain that these factors affected patients' preparedness and readiness to accept ART when offered. They reported that due to the surprise surrounding this diagnosis, patients typically deny their positive HIV test results due to the limited knowledge about HIV and the perceived stigma around HIV. They also highlighted that patients without good family social support were unlikely to start ART during hospitalization.

> *"Some patients are not willing to start the ARVs at that time. They want to seek a second opinion before they start the ART. Some of them want to first be given time and they think through it since it is a lifelong treatment. Some patients are not ready, they are hesitant to start on it there and then"*

> *(Internal Medicine specialist trainee)*

## Healthcare worker related factors: Knowledge, limited human resource, and work overload

Inadequate practical knowledge on the approach and benefits of early ART initiation among the very sick hospitalized patients was highlighted as a barrier. Many healthcare workers lacked refresher training about current ART guidelines, lacked the confidence to initiate ART partly due to the previously observed experience of ART related toxicities among sick patients while others especially the junior doctors, did not consider ART as an emergency.

> *"Some healthcare workers just hear about the HIV test-and-treat but they do not know what entails in that policy, so there is a lack of knowledge about it. And then the other important thing is that sometimes there is 'inertia'. You diagnose somebody and you just say that let me start on them (ARVS) next week."*

> *(Internal Medicine specialist trainee)*

Healthcare workers (10/12) felt the workload was too much for them to offer comprehensive HIV care for hospitalized patients. While the counselors and ART nurses on the ward reported a lack of sufficient time to prepare patients for ART initiation, the clinicians noted that they had many other patients on the wards to attend to.

> *"If it is only one person talking to many people sometimes you get tired and you miss out some information and someone does not get satisfied and he will say that I will not take ART"*

> *(ART nursing officer)*

## System related factors: Pre-ART initiation laboratory workup, linkage facilitators, ART availability

Nearly all HCWs noted that health facilities do not offer ART services over the weekend. This implied that if a decision to start ART was made during the weekend, patients had to either

wait till Monday or be discharged without initiating ART. The majority of patients opted to be discharged without ART promising to return to the outpatient clinic. There were no linkage facilitators to ensure all patients newly diagnosed and opting to start ART from a different center were not lost to follow up.

> *"We should try as much as possible to have those services 7 days a week because you find that patients who come on weekends will have to wait until Monday to have an HIV test done, so they will have missed two or three days while they are in hospital, and also the other thing is the availability of drugs because you find that if you happen to do the test on weekend and the patients is really positive, drugs are not available."*

> *(Intern medicine specialist trainee)*

The lack of consistently available free laboratory testing services in public health facilities was highlighted as one of the commonest reasons for delayed ART initiation in the hospital. Patients having to incur costs to do laboratory tests deemed necessary prior to starting ART including liver and kidney function was a hindrance since the majority of patients had limited financial capacity to pay for these tests. This contributed to delays in ART initiation or missed opportunities to initiate ART during hospitalization.

> *"Unfortunately most of the patients who come, who are "ISS" (HIV-infected) tend not to have money so that's where it goes strange, you get constrained. If the patient is a little clinically stable, it means you have to send them home, to come back for review in the clinic with some of those labs, so mostly you find that they end up being initiated in the outpatient clinics."*

> *(Intern Doctor)*

## Discussion

In this prospective mixed methods study, we found that just over half (52.6%) of the hospitalized patients recently diagnosed with HIV infection initiated ART during hospitalization and only a minority (28.5%) initiated ART within 24 hrs of admission and testing. The majority (64%) presented with advanced HIV disease. The overall all-cause 30-day mortality was 14.5%, and most deaths occurred during hospitalization. Barriers to ART initiation were mainly related to the complexities of dealing with very sick patients who presented with advanced HIV disease and poor in-patient pre-ART preparation.

To our knowledge, since the rollout of the HIV test-and-treat policy no other prospective studies in Uganda have estimated ART initiation specifically among hospitalized patients recently diagnosed with HIV infection. In our study, the proportion of patients with a new HIV diagnosis who started ART while hospitalized (52.6%) was comparably higher than previously reported in a retrospective study in Mulago National Referral Hospital, however, the criteria to initiate ART was notably different in the two studies. Only 5.7% of ART eligible patients had initiated ART during their time of hospitalization and only 20% had initiated ART within 2 weeks in the retrospective study [10]. These high proportions we found are likely explained by the change of ART initiation practice among HCWs following the adoption of new national guidelines since 2016, recommending immediate ART initiation irrespective of CD4 count [5].

Our study also found that only 28.5% started ART within 24 hours of admission or the day they received their HIV test result (same day initiation) and 91.9% had started within 2 weeks,

this finding is comparable to retrospective studies conducted in sub Saharan Africa, which assessed ART initiation among PLHIV in out-patient and in-patient settings [15,16]. In a retrospective study conducted in South Africa reviewing routine data since the rollout of HIV test and treat in 2017, the proportion of PLHIV that had initiated ART on the same day was 30.3% in 2017, however, this increased to 54.2% in 2018 [15].

Overall rapid ART initiation remains relatively low among hospitalized adults, despite free access to ART in most low and middle income countries, however, among special groups including pregnant women, the proportion starting ART on the same day is higher than adults in the general HIV population in the same age group [17]. These low proportions of same day ART initiators among hospitalized patients from our study is likely due to interacting factors. First, the proportion of patients admitted with advanced HIV disease was high, this is similar to what other studies in LMICs have found [18–20], the current HIV treatment guidelines The current WHO guidelines for initiation of ART among patients with HIV suspected to have TB should initiate ART as soon as possible. In those with confirmed TB, ART should be initiated as soon as possible within 2 weeks of starting Anti-TB drugs. Exception is made in patients suspected to have TB or cryptococcal meningitis; ART should be delayed at least until 4 weeks after initiating treatment [4,6,20,21]. Second, the complexities in managing patients with advanced HIV disease creates clinical challenges for healthcare workers making it difficult to decide to initiate early ART [22–25], this was partly supported by the significant proportion of patients (27.3%) who died before starting ART or the attending clinician felt they were too sick to start ART. Third, the challenges in obtaining pre-ART laboratory workup including test stock outs and long turnaround time affect rapid ART initiation in hospitalized patients. Finally, a significant proportion (22.7%) of ART eligible patients opt not to start ART when it is offered and this is partly attributed to system factors like lack of privacy in the open model hospital wards, limited human resource and thus limited time spent interacting with the patient, this negatively impacts the patient-healthcare worker ART initiation preparation process, thus affecting their readiness to initiate ART [26–28]. Patient factors like lack of formal education, influences their awareness and knowledge about benefits of early ART initiation, [29]. Since our study was conducted in referral hospitals, many patients coming from distant areas were less likely to start ART in hospital, the HCW reported such patients request to seek second opinion from health facilities near their homes, or promise to start ART from facilities next to their homes.

The all-cause mortality from our study is comparably lower than published studies from similar settings prior to the HIV test-and-treat policy [10,30]. In a retrospective chart review conducted at the National Referral Hospital before the adoption of the test-and-treat, HIV/ AIDS was the leading cause of death, with an attributed case fatality rate of 25%. Our mortality rate is also lower than the 26.3% previously reported in a prospective study among hospitalized patients in a tertiary hospital in Lilongwe, Malawi among ART naïve patients or 32.9% among new ART initiators in which TB and Cryptococcal meningitis had the highest case fatality rates [31].

This is the first study in Uganda since the rollout of the HIV test-and-treat strategy in 2016 that specifically attempts to broadly characterize ART initiation, explore the barriers to implementation among hospitalized patients with advanced HIV disease. We believe our findings could be useful in planning and developing strategies towards eliminating AIDS related deaths. However, the study had some limitations mainly from recruiting participants from referral hospitals that tend to have very sick patients needing complex clinical care. Our methods of measuring post discharge outcomes through phone interviews may not be as robust as physical home visits. Only half of the participants had a CD4 measurment. Although this was an observational study that was not powered to examine the mortality difference between the two

patients groups, we nevertheless found significantly lower mortality among those who initiated ART during hospitalization similar to what has been previously published by the AIDS Clinical Trial Group (ACTG 5164) and the IDEAL-study [32,33]. We cannot exclude the impact of the Hawthorne effect on our study, thus, our results should be generalized to other hospital settings with caution.

## Conclusion

Early ART initiation following the HIV test-and-treat strategy is feasible, however, challenges to adhere to this strategy remain, especially among hospitalized patients with advanced HIV disease who contribute to HIV-related mortality. Inadequate staffing and inadequate healthcare workers' knowledge around the complexities of care for patients with advanced HIV disease further contributes to this missed opportunity to rapidly initiate ART as a strategy to minimize HIV-related mortality.

## Supporting information

**S1 File. STaTa data set.**
(ZIP)

**S2 File. Health care worker interview guide.**
(DOCX)

## Acknowledgments

We would like to extend our sincere thanks to the participants, both patients and the health care workers for sparing time to participate in the study. Much appreciation goes to the research assistants Esther Kasirye, Richard Sebulime and Hawa Namulondo for their efforts towards making this study a reality.

## Author Contributions

**Conceptualization:** Andrew Katende, Lydia Nakiyingi, Irene Andia-Biraro, Fred C. Semitala, David B. Meya.

**Data curation:** Andrew Katende, Richard Muhumuza, Andrew S. Ssemata, David B. Meya.

**Formal analysis:** Andrew Katende, Thomas Katairo, Richard Muhumuza, Andrew S. Ssemata, Christopher Nsereko, Fred C. Semitala, David B. Meya.

**Funding acquisition:** Andrew Katende, Fred C. Semitala, David B. Meya.

**Investigation:** Andrew Katende, Richard Muhumuza, Fred C. Semitala, David B. Meya.

**Methodology:** Andrew Katende, Lydia Nakiyingi, Irene Andia-Biraro, Thomas Katairo, Richard Muhumuza, Andrew S. Ssemata, Christopher Nsereko, Fred C. Semitala, David B. Meya.

**Project administration:** Andrew Katende.

**Resources:** Andrew Katende, Lydia Nakiyingi, Christopher Nsereko, Fred C. Semitala, David B. Meya.

**Software:** Andrew Katende, Thomas Katairo.

**Supervision:** Andrew Katende, Lydia Nakiyingi, Irene Andia-Biraro, Christopher Nsereko, Fred C. Semitala, David B. Meya.

**Validation:** Andrew Katende, Lydia Nakiyingi, Irene Andia-Biraro, Thomas Katairo, Richard Muhumuza, Andrew S. Ssemata, Christopher Nsereko, Fred C. Semitala, David B. Meya.

**Visualization:** Andrew Katende, David B. Meya.

**Writing – original draft:** Andrew Katende.

**Writing – review & editing:** Andrew Katende, Lydia Nakiyingi, Irene Andia-Biraro, Thomas Katairo, Richard Muhumuza, Andrew S. Ssemata, Christopher Nsereko, Fred C. Semitala, David B. Meya.

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
