## [Decision Letter · Decision Letter 0]

28 Oct 2021

PONE-D-21-11016Anti-retroviral Therapy Initiation and Outcomes Of Hospitalized HIV-infected Patients in Uganda - An Evaluation  of the HIV Test and Treat strategy.PLOS ONE

Dear Dr. Katende,

Thank you for submitting your manuscript to PLOS ONE. After careful consideration, we feel that it has merit but does not fully meet PLOS ONE’s publication criteria as it currently stands. Therefore, we invite you to submit a revised version of the manuscript that addresses the points raised during the review process.

Please address issues raised by the reviewers, particular around the first point raised by reviewer 2. Overall,  better contextualizing the  hospital admission population in the result and discussion is also needed.

We look forward to receiving your revised manuscript.

Kind regards,

Nei-yuan Hsiao

Academic Editor

PLOS ONE

Journal Requirements:

"The author (AK) received funding for this work through Makerere University Implementation Science program: Building Implementation Science Capacity at Makerere University to Strengthen the Response to HIV/AIDS Epidemic in Uganda. This work was supported by the Fogarty International Center of the National Institutes of Health under Award Number D43 TW010037. Prof Moses R Kamya, (MRK) recevied this grant.  The content is solely the responsibility of the authors and does not necessarily represent the official views of the National Institutes of Health. The funders had no role in study design, data collection and analysis, decision to publish, or preparation of the manuscript" 

We note that one or more of the authors is affiliated with the funding organization, indicating the funder may have had some role in the design, data collection, analysis or preparation of your manuscript for publication; in other words, the funder played an indirect role through the participation of the co-authors. If the funding organization did not play a role in the study design, data collection and analysis, decision to publish, or preparation of the manuscript and only provided financial support in the form of authors' salaries and/or research materials, please do the following:

a. Review your statements relating to the author contributions, and ensure you have specifically and accurately indicated the role(s) that these authors had in your study. These amendments should be made in the online form.

b. Confirm in your cover letter that you agree with the following statement, and we will change the online submission form on your behalf: 

“The funder provided support in the form of salaries for authors [insert relevant initials], but did not have any additional role in the study design, data collection and analysis, decision to publish, or preparation of the manuscript. The specific roles of these authors are articulated in the ‘author contributions’ section.

Reviewers' comments:

Reviewer's Responses to Questions

**Comments to the Author**

1. Is the manuscript technically sound, and do the data support the conclusions?

Reviewer #1: Yes

Reviewer #2: Yes

2. Has the statistical analysis been performed appropriately and rigorously? 

Reviewer #1: I Don't Know

Reviewer #2: Yes

3. Have the authors made all data underlying the findings in their manuscript fully available?

Reviewer #1: No

Reviewer #2: No

4. Is the manuscript presented in an intelligible fashion and written in standard English?

Reviewer #1: Yes

Reviewer #2: Yes

5. Review Comments to the Author

Reviewer #1: Thank you for the opportunity to review this paper. It is a well-written manuscript that touches on an interesting and relevant topic. I have no major comments or suggestions for revisions. Some minor suggestions/comments:

1. Background: The authors might consider expanding on what we know about Test and Treat- a lot of work has been done to describe uptake and retention with Test and Treat policies (including Option B+). The authors could consider adding some of these details to provide a complete picture of how Test and Treat has been assessed thus far (challenges, successes, etc)

2. Could be interesting in any future work to conduct qualitative interviews with patients- specifically those who decline to start ART at the time of hospitalization. Data from the HCW interviews provide some insights into those who opt to start vs. those who do not. Interviews with patients would provide additional interesting insights.

3. The authors may consider providing details of what the current clinical recommendations are for those who are TB/HIV co-infected. In many clinical settings, there is confusion on this point with HCWs unclear on what the guidelines are or discomfort with shifting guidelines. In interviews with the HCWs- was it clear that they were providing clinical care that was in line with the current national or global guidelines?

Reviewer #2: This is a useful study of an important unanswered question around the timing of ART initiation in hospitalised HIV patients. The study group are medically complex and diverse, making it difficult to draw clear conclusions.

First 2 broad comments:

1. There are clear clinical contra-indications to immediate ART initiation in this group e.g. CCM, TB, critically ill, died as in your Table 3. This group should be removed from your analysis of 'factors associated with early ART'. i.e. The demographics, distance from home etc are only relevant to analysing the patient factors for deferring ART. Otherwise could be confounders. You don't discuss all the factors found significant on your Table 4, and I suspect many were chance findings as there isn't a plausible link that comes to mind. Eg, marital status? Please recheck Education figures, and no CI for employment?

2. In your discussion you have to highlight the fact that this was a pragmatic, observational study with the MAJOR confounder between the 2 groups being that those with TB, CCM and early death weren't initiated (appropriately). You shouldn't be tempted to draw any conclusions about lower mortality in a highly selected less sick group initiated early. It would require a carefully designed RCT to adequately answer this Q.

3. Table 2> What relevance are WBC, PLT, ALT, AST? Suggest omit

4. It would be useful to compare the primary reasons for admission/ final diagnosis between the Early ART & deferred groups if possible to help the reader understand this patient group.

5. "The proportion (95% confidence interval) of hospitalized patients with a new HIV

diagnosis who initiated ART during hospitalization was calculated as the number of

patients initiated on ART out of the total number of hospitalized patients with a new HIV

diagnosis admitted in the hospital during the study period." - Surely it's rather the total number recruited for the study?

Your qualitative aspect does raise some important programmatic issues.

only half having CD4 is a limitation

Thanks

Cl

6. PLOS authors have the option to publish the peer review history of their article (what does this mean?). If published, this will include your full peer review and any attached files.

Reviewer #1: No

Reviewer #2: No

---

## [Author Response · Author response to Decision Letter 0]

5 Feb 2022

Response to Reviewers

Comment: Please review your reference list to ensure that it is complete and correct. If you have cited papers that have been retracted, please include the rationale for doing so in the manuscript text, or remove these references and replace them with relevant current references. Any changes to the reference list should be mentioned in the rebuttal letter that accompanies your revised manuscript. If you need to cite a retracted article, indicate the article’s retracted status in the References list and also include a citation and full reference for the retraction notice.

Response: Thank you for your comment; I have added one reference, 

Kerschberger, B., Jobanputra, K., & Schomaker, M. (2019). Feasibility of antiretroviral therapy initiation under the treat-all policy under routine conditions: a prospective cohort study from Eswatini. 22(10), e25401. doi: 10.1002/jia2.25401

Comment: We note that the grant information you provided in the ‘Funding Information’ and ‘Financial Disclosure’ sections do not match. 

Response: Thank you for your guidance

 I agree with the following statement financial statement; “The funder provided support in the form of salaries for authors [FCS, AK], but did not have any additional role in the study design, data collection and analysis, decision to publish, or preparation of the manuscript. The specific roles of these authors are articulated in the ‘author contributions’ section. This statement has been adapted in the funding source section page 18. 

Comment: Please include captions for your Supporting Information files at the end of your manuscript, and update any in-text citations to match accordingly. Please see our Supporting Information guidelines for more information:

Response: Captions of the supporting information files have been put. 

S1 Table.

S2 STaTa data set

Reviewers’ comments

Comment: Have the authors made all data underlying the findings in their manuscript fully available?  The PLOS Data policy requires authors to make all data underlying the findings described in their manuscript fully available without restriction, with rare exception (please refer to the Data Availability Statement in the manuscript PDF file). The data should be provided as part of the manuscript or its supporting information, or deposited to a public repository. For example, in addition to summary statistics, the data points behind means, medians and variance measures should be available. If there are restrictions on publicly sharing data—e.g. participant privacy or use of data from a third party—those must be specified.

Reviewer #1: No

Reviewer #2: No

Response: Thank you for your guidance

The anonymised quantitative data set has been uploaded as the supporting document. However the qualitative recordings and transcripts contains potentially sensitive information about the health care workers and their respective hospitals, this will be against the respective hospital ethical review boards and the Makerere University IRB. Attached is the contact for Makerere University IRB in case of further requests for qualitative data requests; rresearch9@gmail.com OR research@chs.mak.ac.ug

Comment: Could be interesting in any future work to conduct qualitative interviews with patients- specifically those who decline to start ART at the time of hospitalization. Data from the HCW interviews provide some insights into those who opt to start vs. those who do not. Interviews with patients would provide additional interesting insights.

Response: We thank you for this comment, we also acknowledge this as a limitation and surely patient interviews would provide additional insight. 

Comment: The authors may consider providing details of what the current clinical recommendations are for those who are TB/HIV co-infected. In many clinical settings, there is confusion on this point with HCWs unclear on what the guidelines are or discomfort with shifting guidelines. In interviews with the HCWs- was it clear that they were providing clinical care that was in line with the current national or global guidelines?

Response: Thank you for this suggestion 

The current WHO guidelines for initiation of ART among patients with HIV suspected to have TB should initiate ART as soon as possible. In those with confirmed TB, ART should be initiated as soon as possible within 2 weeks of starting Anti-TB drugs. Exception is made in patients suspected to have TB or cryptococcal meningitis; ART should be delayed at least until 4 weeks after initiating treatment. (Discussion, Page 16) 

Comment: There are clear clinical contra-indications to immediate ART initiation in this group e.g. CCM, TB, critically ill, died as in your Table 3. This group should be removed from your analysis of 'factors associated with early ART'. i.e. The demographics, distance from home etc are only relevant to analysing the patient factors for deferring ART. Otherwise could be confounders. 

Response: Thank you so much for this observation, we have removed TB, CCM from the multivariate analysis of 'factors associated with early ART'. Since Table 3 is rather a descriptive table, showing various reasons for deferring ART we kept them there. 

Comment: You don't discuss all the factors found significant on your Table 4, and I suspect many were chance findings as there isn't a plausible link that comes to mind. Eg, marital status? 

Response Thank you so much for this important suggestion, We have only discussed plausible factors like distance from health facility and education status, Discussion, page 16 and 17

Comment: Please recheck Education figures, and no CI for employment?

Response: Thank you for this observation

There is one patient who did not have an outcome (ART status at discharge), we put a foot note on table 4 (*) clarifying on the n= 110 instead of 111

These figures have now been corrected, thank you so much for the observation

The employment variable has no CI because it didn’t go to the multivariate model, we only provided CI to those that made it to the multivariate model 

Comment: In your discussion you have to highlight the fact that this was a pragmatic, observational study with the MAJOR confounder between the 2 groups being that those with TB, CCM and early death weren't initiated (appropriately). You shouldn't be tempted to draw any conclusions about lower mortality in a highly selected less sick group initiated early. It would require a carefully designed RCT to adequately answer this Q.

Response: We acknowledge the short falls of the study design and we have stated that study was not powered to draw conclusions on the difference in the mortality in the limitation section, on page 17

Comment: Table 2> what relevance are WBC, PLT, ALT, AST? Suggest omit

Response: Thanks for the comment; these have been omitted as suggested 

Comment: It would be useful to compare the primary reasons for admission/ final diagnosis between the Early ART & deferred groups if possible to help the reader understand this patient group. 

Response: Data regarding reason of admission/diagnosis rather than TB and CCM, would be very heterogenic and hard to generalize. Since this was an observational study, many patients had more than one reason of admission collected in terms of presenting symptoms, but also multiple final diagnoses which most of them were clinically decided. This information would be very hard to generalize and this is another limitation of the study.

Comment: "The proportion (95% confidence interval) of hospitalized patients with a new HIV diagnosis who initiated ART during hospitalization was calculated as the number of patients initiated on ART out of the total number of hospitalized patients with a new HIV diagnosis admitted in the hospital during the study period." - Surely it's rather the total number recruited for the study?

Response: Thanks for the comment, we have made the changes as suggested on page 5

Comment: Your qualitative aspect does raise some important programmatic issues.

Response: Thank you so much , it a our honor 

Comment: Only half having CD4 is a limitation

Thank you for this comment, this is acknowledged and reported as a limitation on page 17 

Thank you so much

---

## [Editor Report · Decision Letter 1]

25 Apr 2022

Anti-retroviral Therapy Initiation and Outcomes Of Hospitalized HIV-infected Patients in Uganda - An Evaluation  of the HIV Test and Treat strategy.

PONE-D-21-11016R1

Dear Dr. Katende,

We’re pleased to inform you that your manuscript has been judged scientifically suitable for publication and will be formally accepted for publication once it meets all outstanding technical requirements.

Kind regards,

Nei-yuan Hsiao

Academic Editor

PLOS ONE
---

## [Editor Report · Acceptance letter]

11 Aug 2022

PONE-D-21-11016R1 

Antiretroviral Therapy Initiation and Outcomes Of Hospitalized HIV-infected Patients in Uganda - An Evaluation  of the HIV Test and Treat strategy. 

Dear Dr. Katende:

I'm pleased to inform you that your manuscript has been deemed suitable for publication in PLOS ONE. Congratulations! Your manuscript is now with our production department. 

Kind regards, 

on behalf of

Dr. Nei-yuan Hsiao 

Academic Editor

PLOS ONE